# ‘But Because I Don’t Know About It, That’s Why I Haven’t Done It’: Experiences of Access to Preventive Sexual and Reproductive Health Care for Refugee Women from Iraq and Syria Living in Melbourne, Australia—A Qualitative Study

**DOI:** 10.3390/ijerph22020149

**Published:** 2025-01-23

**Authors:** Natasha Davidson, Karin Hammarberg, Jane Fisher

**Affiliations:** Global and Women’s Health, School of Public Health and Preventive Medicine, Faculty of Medicine Nursing and Health Sciences, Monash University, Melbourne, VIC 3004, Australia; karin.hammarberg@monash.edu (K.H.); jane.fisher@monash.edu (J.F.)

**Keywords:** health literacy, Iraq, prevention, qualitative, refugee women, sexual and reproductive health, Syria

## Abstract

Women from Syria and Iraq constitute two of the largest groups of humanitarian visa entrants to Australia in the past 10 years. Barriers to and enablers of preventive sexual and reproductive health (SRH) for these women are poorly understood. The aim of this study was to establish the preventive SRH care needs and experiences of women from refugee backgrounds from Syria and Iraq living in Australia. A qualitative study using semi-structured interviews was conducted with women from Syria and Iraq living in Melbourne, Australia. Caseworkers assisted with recruitment and volunteer interpreters with interviews. Between 1 December 2021 and 17 May 2022, interviews were conducted in English or in Arabic with a volunteer interpreter. Audio recordings of English dialogue were transcribed verbatim. Reflexive thematic analysis was used to analyse and report data. Eighteen women were interviewed. Six themes were identified: (1) Awareness and knowledge about preventive SRH, (2) Perceptions about the need for preventive SRH care seeking, (3) Self-care and lack of motivation to seek preventive SRH care, (4) Health information seeking, and (5) Barriers to and enablers of preventive SRH care. Complex factors were found to influence access to preventive SRH care. Enhancing educational initiatives, improving accessibility to reliable health information, and addressing structural and motivational barriers are important for fostering better preventive SRH outcomes.

## 1. Introduction

Worldwide, the number of people forcibly displaced both across borders and within countries as a result of conflict, violence, persecution and human rights violations has increased by 50% over the past 15 years. In 2011, 42.7 million people were forcibly displaced, growing to 110 million as of June 2023 [1]. The 1951 refugee convention defines a refugee as someone who, due to a well-founded fear of persecution based on factors such as race, religion, nationality, membership in a particular social group, or political opinion, is unwilling or unable to return to their country of origin [2]. Forcibly displaced people are those who meet the United Nations criteria for being a refugee [3] and those people seeking asylum but not yet been accorded refugee status.

Prior to conflict in these countries, Syria and Iraq had relatively well-developed healthcare systems in urban areas, with public health programs offering maternal and child health services, family planning, and reproductive health care [4]. However, access to sexual and reproductive health (SRH) care was uneven, with rural and underserved regions facing significant disparities in availability and quality. Cultural norms often limited women’s use of SRH services, particularly in conservative communities where discussions about reproductive health were stigmatised, especially for unmarried women, leading to limited awareness and use of these services [5]. The current crisis in Syria and ongoing conflict, sanctions and the rise of the Islamic State in Iraq has severely weakened the SRH care infrastructure, with the destruction or disruption of health facilities, and the exodus of health professionals [6,7].

Thirteen years of civil war in Syria has resulted in unprecedented levels of population displacement. Of an estimated 13 million people displaced from Syria, about 5.6 million have fled to other countries [8]. Similarly, since 2003, more than 5 million people from Iraq have left their country because of violence, occupation, and terrorism and have resettled in other countries [9]. As of March 2023, 1.2 million Iraqis remain internally displaced or are refugees awaiting resettlement. United Nations resettlement agencies have assisted many refugees from Syria and Iraq to resettle in high-income countries, including Australia [10]. Women and girls make up approximately 50% of refugee populations [1].

Prior to the recent arrival of Syrian and Iraqi refugees, established communities existed in Australia, with approximately 16,000 and 66,000 people, respectively [11]. In 2015, the Australian Government announced a one-off resettlement of 12,000 Syrian and Iraqi refugees, targeted as key groups in the Syrian conflict and the Iraq war [11]. Between January 2013 and June 2023, the number of humanitarian visa entrants from Syria and Iraqi resettling in Australia totalled 19,400 and 41,454, respectively, representing the two largest ethnic groups to arrive during that period [10]. Twenty-nine percent of these new arrivals resettled in Victoria [12].

People entering Australia on a humanitarian visa are referred to the Humanitarian Settlement Program, which offers a fee free comprehensive health assessment with referrals to services based on identified health needs [10]. People on the Settlement Program are also supported to navigate health services; manage severe, critical, long term and/or unmanaged health needs; and connect to local community groups [10].

Challenges inherent in migration, displacement, and resettlement contribute to adverse health outcomes, particularly among women [13]. Pre-migration experiences including violence, torture, rape, or exposure to the torture or killing of loved ones have negative psychological and physical health consequences [14]. Displacement poses significant difficulties for women, who often leave their home countries abruptly, experiencing the loss of social, health, and cultural support systems and separation from family members [15]. During transit, women are vulnerable to gender-based violence and exploitation, particularly if unaccompanied by men [16]. According to the UNHCR, reports of conflict-related sexual violence among women and girls forced to flee increased by 50% in 2024 compared to the previous year [17]. Evidence indicates that up to 40% of women in humanitarian settings experience sexual or gender-based violence during displacement, with even higher rates in some regions [18]. Post-migration stress associated with navigating new healthcare systems further exacerbates their poor overall health [19].

Additionally, women from refugee-like backgrounds experience poor SRH outcomes. For instance, compared to non-indigenous native-born Australian women, they experience increased rates of unintended or mistimed pregnancies [20], lower levels of contraceptive use [21], and increased risk of perinatal deaths and other adverse perinatal outcomes [22].

Existing research in Australia among women from culturally and linguistically diverse backgrounds, including refugees, has focused on their experiences using mental health and maternal health care services following resettlement, with the aim of understanding how the health services could be improved [23,24]. Limited research exists examining access to SRH services for Syrian or Iraqi women, which to date has focused on maternal healthcare, contraceptive care, sexually transmitted infections, and sexual health [25,26]. Barriers to accessing services reported by women include lack of transport and childcare, poverty, and social isolation. Meldrum and colleagues (2016) examined the sexual health knowledge and needs of young Muslim women, including refugees. In this study, women reported a lack of culturally and religiously sensitive support and expressed the need for sexual health services to be more anonymous or provide patient confidentiality [26].

A small number of studies in Australia have explored barriers to accessing SRH care for women of different ethnicities, including women from Arabic [27], Bhutanese [28], South Asian and African backgrounds [29]. These studies show differences in barriers to and enablers of SRH care experienced by women of different ethnicities. Less research has explored the experiences of *preventive* SRH such as contraceptive care, cervical screening, human papillomavirus (HPV) vaccination and breast screening of refugee women from Syria and Iraq.

Health literacy refers to individuals’ abilities to acquire, comprehend, evaluate, and apply information to make informed health-related decisions [30]. Although some studies with people from refugee backgrounds have utilised a health literacy framework [31,32], they focus on the process of seeking and acquiring health information [31,32]. There is a paucity of research within the refugee context exploring broader aspects of health literacy, including understanding, evaluating, and applying health knowledge.

Despite the acknowledged significance of health literacy for health outcomes, there is limited research on women’s access to preventive SRH care or their views on the barriers to and enablers of SRH care, using a health literacy framework. Specifically, there is limited understanding of Syrian and Iraqi women’s access to SRH care; how they understand, appraise and apply preventive SRH information; and their perceptions of barriers to and enablers of optimal preventive SRH literacy. The aim of this study was to generate evidence to inform policy and practice about the preventive SRH needs of women born in Syria and Iraq who arrived on Australia through the Humanitarian Visa Program by exploring if and how they access preventive SRH information and services; and the barriers to and enablers of access to preventive SRH care.

## 2. Materials and Methods

### 2.1. Study Design

A qualitative descriptive design was used in this study to gain insights from participants regarding experiences of access to preventive SRH care for refugee women from Iraq and Syria. Qualitative research is inherently context-specific and focuses on depth rather than breadth, making its findings not generalisable to the broader population but valuable for providing rich, detailed insights into specific phenomena. Semi-structured interviews were used to collect data as they are suitable for studying people’s perceptions and opinions on complex or emotionally sensitive topics [33].

### 2.2. Setting and Context of the Study

The study was conducted in Melbourne, Australia. Large populations of Victorian humanitarian arrivals live within the local government area of Hume where the research was undertaken. In 2021, approximately 40% of the population in this region was born overseas and about 49% speak a language other than English at home [34].

According to the 2021 Australian Census, there were 92,922 Iraq-born and 29,096 Syrian-born people residing in Australia [35]. While specific figures for Melbourne are not provided in the available data, historically, a significant proportion of the Iraq-born and Syrian-born population have settled in Melbourne. These communities are concentrated in specific suburbs, such as Broadmeadows, which was selected as a key study site due to the higher density of people with refugee backgrounds from these countries [36].

### 2.3. Inclusion Criteria

Women were eligible to participate if they were: born in Syria or Iraq, aged between 18 and 50, and had resettled under the Humanitarian Program.

### 2.4. Participants and Recruitment

Information flyers about the study’s purpose, eligibility criteria and the researcher’s (ND) contact details were distributed widely in English and Arabic to stakeholder organisations responsible for resettlement of Humanitarian Program entrants. Through these avenues, the researcher (ND) was introduced to the intake case manager for Victorian Arabic Social Services who recruited participants. The case manager contacted eligible women directly by phone and asked if they were willing to be interviewed. Women who wished to participate confirmed their availability with the case manager who arranged an interview. Convenience sampling was used based on their availability to participate. Oral and written informed consent was obtained before the interview.

### 2.5. Conceptual Framework

Sorensen’s Integrated Model of Health Literacy (2012) was the framework used for this study [37]. Health literacy is linked to literacy and entails people’s knowledge, motivation and competencies to access, understand, appraise, and apply health information [37]. Low health literacy and health knowledge contribute to poor health service access for people of refugee backgrounds [38].

### 2.6. Data Source

An interview guide was developed based on findings of a systematic review [39], the components of Sorenson’s Integrated Health Literacy Model [37] and the researcher’s (ND) refugee health experience. Information about access, use, knowledge and sources of information used, and health care experiences across four areas was ascertained: contraceptive care, cervical screening tests, HPV vaccination and breast screening.

### 2.7. Data Collection Procedure

Davidson and colleagues (2022) conducted a systematic review [39] and identified themes relating to SRH care access including knowledge, awareness, and perceived need for and use of preventive SRH care and language and communication barriers. This evidence directly informed the interview questions; “Have you heard about women’s health problems? [contraceptive care, cervical screening, HPV vaccination and breast screen], “Have you had a… [cervical screening, HPV vaccination and breast screen]” and “Do you know why it’s done [cervical screening, HPV vaccination and breast screen]”?

Sorensen’s Integrated Health Literacy Model directly informed the interview questions; “How do you find your way to health care?”, “How do you find health information?” and “What is the best way to get the health information you want about women’s health?”

Sociodemographic information including age, year of arrival in Australia, level of education, religion, relationship status, number of children, employment status and occupation was collected at the beginning of the interview. The full interview guide is available in Appendix A.

During 1 December 2021 and 17 May 2022, face to face and telephone semi-structured interviews were conducted by ND with the assistance of a volunteer interpreter. Volunteer interpreters were native Arabic speaking higher degree research students experienced in qualitative research methods. They also had experience of the ethics of conducting research with participants from culturally and linguistically diverse backgrounds. Interpreters were not considered part of the research team. Participants were assured their responses would be anonymous. Data collection continued until adequate data were generated to address the research question [40].

### 2.8. Data Management and Analysis

Interviews were conducted in English with a volunteer interpreter explaining questions in Arabic and participant responses translated from Arabic into English during the interview. Participants responded in English, depending on their English language skills. In cases where women did not understand the interview questions, the interviewer directed the interpreter to rephrase the question in simpler terms or to use culturally familiar examples to convey the intended meaning without altering the core of the question. Interviews were audio recorded and the English language dialogue later transcribed verbatim by ND. Quotes presented are translations provided by the volunteer interpreter. Data management and coding were undertaken using NVIVO 12 software.

Reflexive thematic analysis is a method for identifying, analysing, organising, describing, and reporting themes identified within the data and by incorporating reflexivity into the analytical process [40]. Data were analysed using this method [40]. A combination of both deductive and inductive approaches was used for coding. Pre-defined codes were generated based on questions in the interview guide. Throughout the analysis, the coding underwent several revisions as themes were identified on the basis of the responses. Throughout the revisions, ND crossed-checked the coding with JF and KH to ensure consistency within and between interviews. Coding was performed independently by ND for each transcript by close line by line reading. JF and KH reviewed a sample of the coded transcripts for consistency. No major disagreements between authors over theme identification took place. Once this process was complete, patterns across interviews were identified and presented as themes and subthemes.

In reflexive thematic analysis, reflections on cross-cultural factors and how the interviewer’s perspective influence the research process are important, especially in conveying sensitive concepts across languages and cultures. When conducting interviews and translating into Arabic, the nuances of certain concepts about preventive SRH were considered in the translation. Concepts and questions were adapted to align with the cultural norms and values of the participants. In addition, the interviewer’s cultural background, beliefs, and experiences shaped her worldview, which influenced how questions were framed. During the thematic analysis, the interviewer’s perspective also affected how data were interpreted, how they were identified, how the data were categorised, and how narratives were constructed. To mitigate these influences, the interviewer maintained a reflexive journal to document her thoughts, biases, and reflections throughout the research process.

This study was approved by the Monash University Human Research Ethics Committee (Project ID 28080) on 15 June 2021.

## 3. Results

Eighteen women were interviewed. Ten interviews were conducted face to face and eight over the telephone. Seventeen were conducted in a private room either in person or by telephone at a local community centre and one in the participant’s home. The interview length ranged from 22 min to 1 h 25 min with an average length of 48 min. Sociodemographic characteristics are shown in Table 1 and Table 2.

Six major themes and seventeen subthemes were identified and are summarised in Table 3.

### 3.1. Theme 1: Awareness and Knowledge About Preventive SRH Care

Awareness and knowledge of preventive care varied by SRH topic. From their home country experience, most women were aware of breast screening and contraception and to a lesser extent cervical screening: *“I know about it* [contraception] *from Iraq, from the doctors in Iraq”* (Milad, 31). However, no women had heard about HPV vaccination: *“It’s the first time I’ve heard about this vaccination, I don’t have any idea about it”* (Sara, 31)

About half the women had not heard of cervical screening and had not been screened since arrival in Australia. Of the women who were aware, cervical screening was still not undertaken:


*“I did hear about cervical screening, I didn’t do it. I don’t know for what they do the cervical screening for” (Habda, 57)*


This point is further supported by some women’s suggestion that knowing about SRH heath is not incentive enough to seek care. As Milad states “*Maybe she* [her mother] *is educated but she is careless, they might know about the knowledge* [of cervical screening] *but don’t care”* (Milad, 31)

#### 3.1.1. Understanding of Screening and HPV Vaccination

Of the women who knew about cervical screening and breast screening, half believed it was performed to prevent cancer. The remaining half did not understand the reason for screening:


*“Actually, I had experience in Syria before I arrived here in Australia, I did it [cervical screening] in Syria but I don’t have any more information about how important it is compared to now. [In Syria] I just did it because it was a routine thing but I didn’t have any information about why it’s done“ (Basima, 51)*


All but one of the participants were unaware of the link between the HPV vaccine and cervical cancer and did not know the vaccine prevents cervical cancer. *“No, I haven’t heard of it and don’t know why it’s done”* (Lina, 31). This response is indicative of women’s understanding:


*“I know that the school gives vaccinations but I don’t know what for, I only know now when you told me. I think my oldest daughter she might have had it.” (Akhalas, 49)*


#### 3.1.2. Factors Impacting HPV Vaccination Uptake

Lack of available information and beliefs about side effects were factors negatively impacting women’s decision-making regarding HPV vaccination. This response was typical of women when asked “What encourages parents to get the HPV vaccine for their children and what prevents or stops them?”


*“The other things that would prevent parents from getting their children vaccinated would be they are afraid or worried their children would have any side effects. In our community we don’t know about that vaccine, its new to us. What is that vaccine, it’s the first time I’ve heard about it, maybe it causes something for my children, maybe causes hormone changes that’s what I think.” (Sara, 31)*


For a minority of women, misguided beliefs were responsible for reducing HPV vaccine uptake:


*“Some people here in my community are afraid, they take a lot of vaccination here in Australia, they take more than one maybe that’s not good for our health” (Zeinah, 51)*


### 3.2. Theme 2: Perceived Need for Prevention

#### 3.2.1. Seeking Care When Visible Symptoms Were Present

Over half of the women knew screening was performed to detect cancer *“or to see if there are any diseases and everything is normal”* (Samar, 31) in the breast or cervix. Yet for a minority, care was sought only when women exhibited symptoms or when their immediate relatives such as their mother or aunt experienced disease.


*“I have heard about it, I felt a pain in my breast the first thing when I arrived in Australia, I was lucky I went to the doctor and we found this lump in the early stage and so we did the surgery and its removed now… and she has been in treatment for 4 years now.” (Nana, 46)*


#### 3.2.2. Positive Influence of Other Women in the Community

The experience of relatives in women’s communities was a strong incentive for breast screening. Reem outlines her experience:


*“Because like my sister felt a little bit of a lump in her breast. She went to the doctor, the doctor said you should make a breast scanner and X-ray… When my sister went to the doctor she said to me and my mum and my other sister, you should go to do the X-ray of your breast” (Reem, 50)*


#### 3.2.3. Limited Perceived Need for HPV Vaccination

Women held conflicting views regarding decision-making about HPV vaccination. In some cases, women did not believe HPV vaccination was relevant to their daughters due to young women’s restricted sexual activity before marriage. A minority of women were largely unaware of the vaccine or that it protects against cervical cancer.

Women expressed a lack of knowledge, low health literacy, and inadequate information provision about HPV vaccination. Of the women with high school aged children eligible for HPV vaccination, most had not heard about it and were not aware if their children had received the vaccine. Of the few women who had heard of it, a minority stated they would be uncomfortable particularly with their female children having the HPV vaccine. It was strongly held belief that unmarried girls are not sexually active before marriage so there is no perceived need for HPV vaccination. This participant explains:


*“I think that the girls don’t need to take this vaccine until they get married from my understanding and why they [parents] prevent them because they know their children and they know the way they brought them up, they will not have any relationship before they get married” (Akhalas, 49).*


Conversely, a minority of women when asked “If they did know about it and it was offered” were more accepting of the vaccine. *“If I know about it* [HPV vaccination] *of course I would give my son and daughter.”* (Habda, 57) *“I wish there was more information about this topic so I can be aware about it”* (Zeinah, 51).

Several women suggested that low perceived need for HPV vaccination stemmed from women’s own country of origin experiences.


*“There are a lot of people when we were young they didn’t take these vaccines so why are we taking it now. We didn’t get anything then so that’s OK so they don’t encourage their children to have it because they [parents] never had it when they were young. Also like with COVID vaccination people don’t trust the vaccine maybe there is something bad in it, some people do trust it and they had it. [COVID vaccination]” (Wala, 32)*


#### 3.2.4. Negative Social Influences on Contraceptive Use

Overall family and community had a significant impact on most women’s contraceptive access and use. Women’s beliefs regarding the ineffectiveness of contraception is reflected here:


*“Because of some experience with her friends they put something in her uterus [IUD-Helix] you know. And that didn’t prevent them from having a baby. They used an IUD and they didn’t help them they also had a baby after the IUD insertion. For that she doesn’t trust the medication or that way to stop having a baby. Not being effective” [form of contraception] (Zinah, 39)*


Several women believed contraceptive use negatively impacted their opportunity to become pregnant.


*“I had difficulty getting pregnant so I am against having contraception, I love having kids. I don’t agree with contraception” (Milad, 31)*


### 3.3. Theme 3: Self-Care and Motivation

#### 3.3.1. Low Prioritisation of Preventive Health Care

Women indicated they did not prioritise screening tests. When asked why other women in their communities do not undertake cervical or breast screening, several women cited competing priorities such as family and caring commitments taking precedence over seeking preventive SRH care: *“They* [women] *feel it is not important, it is not necessary to do that in their in life…”* (Lina, 31) and believed that *“If I am busy and have a lot to do, because I am a women you just think kids family, family, family”* (Samar, 31)

#### 3.3.2. Fear and Embarrassment Undergoing Screening

Fear of pain during the screening procedure, the possibility of negative findings and embarrassment undergoing screening were common deterrents to screening.


*“In Australia what prevents the women in her community from doing that maybe if they feel afraid from that or they don’t like to feel any pain.” (Zinah, 39)*



*“…in our community there are many women who feel ashamed or scared or shy to do that even if they have experienced pain in their breast and sometimes this [breast screen] will make the situation worse” (Habda, 57)*


While most women believed breast screening was physically uncomfortable, some agreed the discomfort was tolerable and this would not prevent them from screening. A minority were still motivated to get a breast screen now or in the future as they were aware of the benefits of screening:


*“I don’t like it, it wasn’t comfortable but I had to do it. When they do the X-ray they press the breast and it was very uncomfortable… the only thing was that I was very terrified, just I want to make sure I’m in good health.” (Zienah, 51)*


Over half of the participants related that women were afraid of finding out about the terminal nature of the disease related to the breast and cervical screening procedure and this prevented them from undertaking screening. This was a common sentiment.


*“Yes, they are scared of what they might find, they might find cancer and they are going to have their breast removed and maybe they will die.” (Milad, 31)*


### 3.4. Theme 4: Health Information Seeking

#### 3.4.1. No New Knowledge Is Required

Despite their limited understanding of preventive SRH care, few women demonstrated a desire for new knowledge regarding contraceptive care, screening or HPV vaccination. When asked “Are there things you find confusing and you’d like to know more about with regarding to …?” Overall, women typically responded with “*No there is no need* [for more information], *I know enough information* [about cervical screening]” (Donna, 36) or *“I have a good idea about this”* [breast screen] (Donna, 36) or *“I don’t have any questions about this vaccination”* (Zinah, 39)

Conversely, a minority of older women were interested in seeking health information about breast screening, which was driven by previous cancer experiences of female relatives. For example:


*“I know that from the women in my community we have to do it because if there is something they can detect earlier you don’t have to go through all the stages of care and its very important for us to prevent anything happening to us. Because my aunty came here they [Breast screen] always send her mail the reminder to do the screening and she threw it in the rubbish all the time and she didn’t care about health care and to go to the doctor and do the screening and everything, until she got the cancer, stage 4 intestinal cancer. So now we have her as a lesson to go to the doctor and to do the check-ups.” (Basima, 57)*


#### 3.4.2. Trusted Sources of Information

Friends and family such as mothers, sisters and cousins were primary sources of health information and often women’s preferred sources. *“Like a group discussion or friends, they know… That gives you all the information you need.”* (Wala, 32). Other trusted information sources included GPs and the internet. These information sources were perceived as reliable and useful:


*“… in my country [Syria], I have a lot of information, I like the education, I need a lot of information about this and I searched/researched about this everywhere in google, in the community, with my doctor, with asking the nurse” (Reem, 50)*


Two women described how they would seek their own health information if they considered it necessary or worthwhile:


*“No, so if I agreed with and supported this vaccine, I would go into it and read and do my own research into it. But because I don’t support this vaccine I don’t want to have it and I don’t feel comfortable to give her daughter or son this is why I stopped getting more information” (Shan, 39)*


#### 3.4.3. A Comprehensive Approach to Information Delivery

Women preferred a variety of modes of information delivery, including in person information provided by GPs, education sessions, community groups, online audio-visual material and through media outlets. The majority of women reported a preference for face to face information from GPs in oral and written formats. Of those, most suggested material be available in Arabic and English and presented orally given women’s limited literacy skills. Improving awareness of preventive SRH care through group education sessions in community settings was also helpful. Most women suggested presentations be led by HCPs and facilitated by interpreters:


*“Through community women’s gatherings. Like here in this centre [Victoria Arabic Social Services Centre] they can ask the doctor to come here and talk about these types of topics” (Basima, 50)*


### 3.5. Theme 5: Barriers to Preventive Sexual and Reproductive Health Care

#### 3.5.1. Sociocultural Factors and Unacceptability of Services

Cultural and religious factors act as barriers to women’s access to preventive SRH care. Almost all women commented on the importance of upholding norms of sexual abstinence before marriage and social values emphasising virginity until marriage for women and girls. Overall, women reflected on the social imperative of virginity, which inhibited unmarried women and girls from obtaining cervical cancer screening, HPV vaccination, or contraceptive care. This was demonstrated by this comment:


*“In Iraq single women, not married women they are not allowed to do it [cervical screening], they will refuse to do it if they are unmarried because the woman is a virgin and so no one is touching this private area. So not in Iraq, even in the Arab community women is not allow to touch this area before they get married because they are virgins.” (Arok, 56)*


Despite knowing about it, some women did not undertake cervical screening prior to marriage as the belief was it is unnecessary due to their virgin status: “*Yes I know about that one* [cervical screening] *and when I came to Australia I received the mail to do the cervical screening but I didn’t do it because I am a virgin*” (Sara, 31)

The cultural imperative condemning premarital sex also influenced women’s beliefs about the need for HPV vaccination:


*“I think that the girls don’t need to take this vaccine until they get married from my understanding and why they prevent them because they know their children and they know the way they brought them up they will not have any relationship before they get married” [vaccination]. (Akhalas, 49)*


Husband opposition was reported to be a significant barrier in accessing information about contraception. When asked why women in her community would not take contraception, Reem responded:


*“I met some women from Arabic background, they talk about this topic [contraception]. One woman said I cannot go to the community or go to the doctor or talk about this topic because my husband maybe he will kill me because it’s a shame in my country [Syria] it’s a shame the women to talk about contraception”. (Reem, 50)*


Conversely, some women were more likely to use contraception if there was martial disharmony. This was indicative of some women’s views:


*“Some women because they don’t good life with their husband or their family. He’s coming the baby life is not good, maybe the father is no good, then why is the baby coming. Don’t come with this baby with this father. This father is no good [reasons for using contraception]… I think this maybe so why is he coming for this life [the baby], [father has] no money, not good maybe he doesn’t work, the husband. Maybe they are poor, maybe they all fight fight fight all the time maybe this maybe…” (Milad, 31)*


#### 3.5.2. Religious Beliefs

Fatalistic beliefs, such as women’s assumption that everything happens by the will of God, that life and death are matters of destiny, particularly with regard to breast and cervical cancer, was described by the majority of women. As Basima explains:


*“In the beginning I was so scared, they did five times [cervical screening] and then they said there was something about the size, if it’s is not gone we will have to send you to the consultant just in the beginning I was so scared but now I am OK. If something happens to me this is from God or not, I’m accepting it” (Basima, 51)*


In the same vein, the use of contraception was influenced by traditional religious and cultural beliefs. *“My religion and culture and value supports having kids”* (Milad, 31). When asked about reasons for not using contraception, this participant from Iraq clearly explains the impact of religious teachings:


*“To not use it, is for us its “haram” in our country, to do that, its forbidden or proscribed by Islamic Law” (Maria, 22)*


#### 3.5.3. Previous Country of Origin Experiences

All women reflected on their experiences of contraceptive use and access to screening and vaccination prior to arrival and resettlement in Australia. An enabler to accessing preventive SRH care in general but contraceptive care in particular was the experiences of a few women in their home country of Iraq:


*“Because of what we have experienced in Iraq, the war there is not a lot of money and a lack of education so people have mental health problems and they don’t want to bring kids into this world, they want to look after themselves and they want to enjoy their lives so they are going to use contraception. A lot of women they read and they look after their heath, they know what’s going on and how to take care of themselves.” (Milad, 31)*


Out of pocket financial cost was cited as a barrier to accessing screening in Iraq and Syria. Consequently, perceived cost may act as a barrier to accessing care on resettlement in Australia: *“In Iraq I can’t do it* [cervical screening] *because it’s very expensive and that’s what prevented me from doing it”* (Zinah, 39) and this *“Maybe because it’s* [breast screen] *expensive in our country* [Syria], *that prevents women from doing it* [in Australia]” (Afifa, 39)

#### 3.5.4. Lack of Health Care Provider Endorsement

Women did receive health information on arrival through resettlement services and were facilitated to see a GP. However, during these visits there were missed opportunities to address SRH as less emphasis was placed on women’s preventive SRH care, as evidenced by such comments as:

“When first we arrived here to Australia they [resettlement services] give us all the information, they make a seminar for us and talk about what information, where we can have the health care and they also give us flyers and everything. They give us like all the information and we went to the doctor but they talked about general health they didn’t be specific about women’s health or what we need.” (Habda, 57)

A typical explanation as to why women had not undertaken cervical or breast screening despite seeing GPs since arrival in Australia: *“No I haven’t heard about it, just now I have heard about it. My GP didn’t mention that*… *In the future I’m going to do the test. But because I don’t know about it, that’s why I haven’t done it.”* (Luma, 50).

#### 3.5.5. Communication and Language

For a small minority of women, it was acknowledged that visiting an English-speaking doctor with no interpreter did create barriers to expressing their health needs clearly:


*“Yes, it’s a little bit difficult I understand what he says but I can’t talk or speak freely to reach my idea or problems so there is some miscommunication so I can’t express herself what I need [without an interpreter]” (Afifa, 39)*


### 3.6. Theme 6: Enablers of Preventive Sexual and Reproductive Health Care

#### Health Care Provider Characteristics

Overall, women suggested that language was an enabling factor impacting their communication with the HCP. Almost all women sought care from Arabic speaking doctors and valued the common language, which lead to high satisfaction with care. The following was a typical comment:


*“Yes I feel happy and also when also the doctor speaks my language that makes me feel happy and comfortable. Also, I can know what I want and I am more comfortable to explain what I want. More, happy because I speak the same language” (Zinah, 39)*


## 4. Discussion

This research revealed that women with refugee backgrounds from Syria and Iraq, living in Australia, have limited access to preventive SRH care, in part attributed to socio-cultural, religious and pre-migration barriers. This study also found there were missed opportunities for women to discuss preventive SRH care with their HCPs. Overall, women were not motivated to seek health information and did not prioritise preventive SRH self-care due mainly to completing priorities and lack of perceived need for preventive SRH care.

Despite women being aware of preventive SRH services as a consequence of their home country experiences, this did not always translate to action in seeking services once in Australia. The term ‘apply’ described by Sorensen refers to the ability to communicate and use information to make a decision to maintain and improve health. Our study revealed negative attitudes towards contraceptive use, screening and HPV vaccination were affected by wider sociocultural and religious factors influencing women’s uptake and use. Cultural taboos surrounding the female social imperative of virginity and prohibition of sexual activity prior to marriage have previously been reported as barriers to SRH care in studies of women from diverse ethnicities in resettlement countries [41,42], including Australia [43,44]. Sociocultural factors are well-established barriers to undergoing cervical screening [45] and using contraception [46,47]. For women in our study, cervical screening and HPV vaccination have been seen as a threat to the virginity imperative and thus not supported practices for unmarried women. One possible reason for this might be that in many Muslim societies, premarital sexual activity is not assumed, which significantly influences public health practices [5]. As a result, HPV vaccination is often administered closer to the time of marriage, and Pap smear screening is typically conducted during prenatal care. By contrast, in Australia, the HPV vaccination schedule differs from that of Syria and Iraq and is given at the commencement of high school, while cervical screening tests are recommended from age 25. Another strong influence on women’s access to contraceptive care after marriage was their husband’s demand for children and a lack of spousal support, also supported in other studies [47].

In our study, religious beliefs had a considerable influence women’s views on cervical cancer and breast screening, as women saw these diseases as fatalistic. This was evidenced by women’s assumptions that everything happens by the will of God, and cancer is fated. Fatalistic beliefs regarding disease progression are known to contribute to lower screening rates in various Arabic speaking ethnic groups [48]. This belief in predestination may also contribute to our finding that women emphasised symptoms-based health seeking rather than prevention. In fatalist religious beliefs, there may be a tendency to accept illness or disease as predestined rather than preventable [49]. As demonstrated in our study, fatalist beliefs may have led to a passive approach to accessing care where women did not actively seek preventive SRH measures but instead waited until symptoms manifested, believing the outcome is beyond their control.

Women held conflicting views regarding HPV vaccination. One reason for differing views may be that women did not have accurate information about HPV vaccination, its purpose, and effectiveness in preventing cervical cancer as found elsewhere [50]. Women in our study expressed their lack of knowledge and low health literacy on this topic, and inadequate information provision by HCPs about HPV vaccination. Conversely, we also found if the offer of HPV vaccination was made and encouraged by providers and women were informed about it, vaccination would be more likely accepted as worthwhile. HPV vaccination health education materials provided and discussed by health professionals have been previously proven critical for parents from culturally diverse backgrounds in their decision to vaccinate [51].

Edwards and colleagues (2013) introduced the notion of distributed health literacy, suggesting social support factors positively impact health literacy, consequently fostering engagement in healthcare activities [52]. However, our findings suggest information gained through distributed health literacy may have contributed to poor decision making with regard to HPV vaccination and contraceptive care. Women were negatively influenced by others’ beliefs or perceptions that contraception was ineffective. If information provided by these informal networks is not evidence-based or accurate, women from refugee backgrounds may make decisions about their SRH care based on misinformation or myths, which can lead to inadequate uptake of preventive SRH care [53]. Taken together these findings suggest women from refugee backgrounds rely on their sociocultural and religious beliefs and social connections, particularly if there is limited opportunity to access preventive SRH information from their GPs.

A lack of GP endorsement and missed opportunities to discuss preventive SRH care was identified in our study when women presented for other health concerns. Opportunistic recommendation of screening tests and HPV vaccinations by HCPs are key facilitators for women to undertake these preventive measures [54]. One reason for our findings might be that women preferred and sought care from Arabic speaking GPs. Despite women overcoming language challenges by consulting GPs who conversed in their native language, on the whole they did not receive information about preventive SRH care. Although overseas trained doctors undergo medical training in Australia to meet the country’s standards and requirements for medical practice [55] and have received comprehensive training in preventive health care during their medical education, there remain barriers to providing preventive SRH information.

Sorensen and colleagues (2012) define ‘understanding’ as the capacity to comprehend health-related information and ‘appraising’ as the process of interpreting, filtering, judging, and evaluating information [37]. Overall, our study revealed that women’s family, social, and network connections negatively impacted their acceptance and understanding of contraceptive care and HPV vaccination, and to a lesser extent, breast and cervical screening. Hence, the role of GPs and primary care nurses in providing accessible and accurate health information and improving the health literacy of women is essential. The way in which women understand and appraise information during a consultation is likely influenced by HCPs perceptions of their health literacy [56]. Wittenberg and colleagues (2013) found some HCPs often overestimate their patients’ health literacy, are least comfortable identifying low literacy patients, and assessing a patient’s health literacy level, thereby not fully meeting their patients’ health literacy needs [57]. Taken together, these points highlight important gaps in the provision of care to improve the uptake of preventive SRH measures among this group of women.

## 5. Strengths and Limitations

There were strengths and limitations to this study. Conducting interviews with women from refugee backgrounds, ensuring their ability to express their experiences and have their voices acknowledged was a strength of this study. Interviewing in English with an Arabic interpreter enabled women with limited English proficiency to engage in the subject matter without language barriers. Offering participants proficient in English the choice to be interviewed in their native language or in English allowed them to select the option they felt most comfortable with. These adaptable approaches were essential given the sensitive nature of the topics and to demonstrate cultural sensitivity.

However, there were limitations. Despite employing various strategies to engage community members, including professional networks and word of mouth, these efforts did not effectively reach individuals who were genuinely considered “hard to reach,” such as women not actively involved in services or lacking connections to community networks. The majority of recruited women became aware of the study through direct contact with the community leader, indicating that participants typically had some form of affiliation with local services or community groups. Enhancing outreach activities and networks could potentially facilitate reaching a broader spectrum of women beyond those already engaged with health and community services.

Conducting research with people who have limited English language skills and different cultural backgrounds poses challenges including ensuring the meaning of participants responses are ascertained accurately. This carries the risk of misinterpretation or loss of potentially relevant information during the interpretation process. To address this risk, any confusion that arose in the interview was checked for accuracy by clarifying with the interpreter during the interview and debriefing with the interpreter following the interview to further explain any misconceptions.

## 6. Clinical Practice and Research Implications

The six themes identified in this study highlight critical areas for improving preventive SRH care. Table 4 provides actionable recommendations for each theme.

An effective approach to enhancing comprehension of preventive SRH measures involves optimising interactions between HCPs and patients by clarifying women’s understanding of preventive SRH. Health care providers require a good understanding of the concept of health literacy to support them in improving the health literacy of the women from refugee backgrounds to ultimately promote active participation in their own preventive SRH care, understanding and uptake [58]. Using the teach-back method, an interactive communication and engagement strategy, empowers HCPs to prompt women to reiterate in their own words the information they have received [59]. This method may facilitate exploration of previously acquired information and the practical application of health-related knowledge. For example, highlighting the availability and benefits of HPV vaccination in particular and its role in the prevention of cervical cancer. Several challenges have been identified when applying teach-back in interpreter-mediated health care appointments associated with differing cultural nuances and cultural practices [60]. As a result, the teach-back approach may need to be modified when communicating with women from refugee backgrounds.

At a clinical level, developing culturally responsive practices could include supporting HCPs to inquire about women’s prior healthcare and pre-arrival experiences in Syria and Iraq. Understanding women’s expectations, previous experiences of barriers to access, and awareness of contraceptive care, cervical screening, and HPV vaccination for women after resettlement would enable HCPs to provide more culturally safe care. Future research determining the barriers to and enablers of providing SRH care to women from refugee backgrounds from an HCP perspective is needed to inform health policies and practice for this group.

Future studies could investigate how refugee women collect preventive SRH related information online, focusing on the platforms and resources they frequently use. Research might examine how these women assess the reliability of online SRH information, including the criteria they use to determine trustworthiness. Further research is needed to explore participants’ experiences with conflicting online information and their strategies for deciding which information to trust and follow. Studies might also explore whether SRH information accessed online is available in Arabic and how language barriers affect their understanding and use of the information.

## 7. Conclusions

This research demonstrates that women from Syrian and Iraqi refugee backgrounds experience unmet preventive SRH needs, primarily due to inadequate knowledge of and limited perceived need for preventive SRH care. The health literacy framework guiding this research allowing the researcher to identify how knowledge, access, and communication influenced the women’s experiences, ultimately highlighting key factors that affect their ability to navigate and utilise preventive SRH care services. Understanding their pre-arrival experiences, expectations of care during resettlement, and sociocultural, religious, and community influences can help HCPs and policymakers deliver more culturally safe and targeted care. Additionally, opportunistic discussions about preventive SRH care are essential to better inform these groups of women and improve their access to care.

## Figures and Tables

**Table 1 ijerph-22-00149-t001:** Participants’ sociodemographic characteristics.

	Women (*n* = 18)
** *Country of birth* **	
Iraq	10
Syria	8
** *Age group* **	
20–29	1
30–39	9
40–49	2
50–59	6
** *Marital status* **	
Single	2
Married	11
Widowed	1
Divorced	4
** *Maternal status* **	
No children	5
Children	13
** *Level of completed education* **	
Minimal primary education	2
Primary education	3
Secondary education	11
Tertiary/university	2
** *Employment* **	
Employed	2
Home duties	16
** *Years in Australia* **	
1–5	10
6–10	5
11–15	2
≥16	1

**Table 2 ijerph-22-00149-t002:** Individual participant characteristics (*N* = 18).

Participants (Pseudonym)	Age (Years)	Years in Australia	Occupation	Number of Children	Level of Education Completed	Religion
Maria	22	12	Support worker NDIS *	0	Year 12	Muslim
Hadba (HB)	57	8	Home duties	0	Year 12	Christian
Arok	56	1	Home duties	6	Year 6	Christian
Zeinah (ZI)	51	6	Home duties	2	Year 12	Christian
Luma	50	8	Support worker NDIS *	1	Year 12	Christian
Nana	46	5	Home duties	1	Year 12	Muslim
Basima (BA)	51	7	Home duties	3	Year 12	Christian
Donna (DA)	36	8	Paid work	0	Tertiary/university	Christian
Zinah (ZI)	39	3	Home duties	5	Year 3	Christian
Afifa (AA)	39	4	Home duties	2	Year 12	Christian
Lina	31	2	Home duties	3	Year 9	Christian
Sara (SAA)	31	2	Home duties	0	Tertiary/university	Christian
Reem	50	4	Home duties	2	Year 12	Christian
Akhalas	49	16	Home duties	4	Year 4	Christian
Samar	31	11	Home duties	3	Year 12	Muslim
Milad (MD)	31	2	Home duties	1	Year 12	Christian
Wala (WA)	32	3	Home duties	4	Year 12	Muslim
Shan (SZ)	39	4	Home duties	2	Year 10	Christian

* National Disability Insurance Scheme.

**Table 3 ijerph-22-00149-t003:** Themes and subthemes.

Themes	Subthemes
Theme 1: Awareness and knowledge about preventive SRH care	Understanding of screening and HPV vaccination * Factors impacting HPV vaccination uptake *
Theme 2: Perceived need for prevention	Seeking care when visible symptoms were present * Positive influence of other women in the community Limited perceived need for HPV vaccination * Negative social influences on contraceptive use
Theme 3: Self-care and motivation	Low prioritisation of preventive health care Fear and embarrassment undergoing screening
Theme 4: Health information seeking	No new knowledge is required * Trusted sources of information A comprehensive approach to information delivery
Theme 5: Barriers to preventive sexual and reproductive health care	Sociocultural factors and unacceptability of services Religious beliefs Previous country of origin experiences Lack of health care provider endorsement Communication and language
Theme 6: Enablers of preventive sexual and reproductive health care	Health care provider characteristics

* Indicates theme related to low health literacy.

**Table 4 ijerph-22-00149-t004:** Themes and related recommendations for improving preventive SRH care for women from refugee backgrounds.

Themes	Recommendations
Awareness and Knowledge About Preventive SRH Care	Primary health care services might conduct culturally tailored workshops to increase awareness and develop multilingual educational materials. Workshops might provide additional support for women by identifying more GPs who speak Arabic.
Perceived Need for Prevention	Health campaigns in Arabic emphasising the long-term benefits of preventive SRH care, such as HPV vaccination, might help shift perceptions. Health care agencies might also train healthcare providers to communicate the importance of preventive SRH effectively.
Self-Care and Motivation	Programs fostering self-efficacy through peer support groups could be prioritised. Additionally, policy makers and health care agencies might introduce digital tools such as mobile apps to track SRH goals and reminders for women’s screening and vaccination.
Health Information Seeking	Strengthening access to trustworthy health information through helplines, mobile clinics, and verified online platforms is warranted. Training health care providers to provide accurate and consistent information during routine visits would also be beneficial.
Barriers to Preventive SRH Care	Health care policy makers might advocate for changes to subsidise SRH services and ensure services are available and easily accessible.
Enablers of Preventive SRH Care	Building partnerships between health care services and local community groups can foster trust and facilitate service delivery. Training healthcare providers to adopt culturally sensitive practices will further encourage service use.

## Data Availability

The data are available upon reasonable request from the corresponding author.

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
