# Peer review of "‘But Because I Don’t Know About It, That’s Why I Haven’t Done It’: Experiences of Access to Preventive Sexual and Reproductive Health Care for Refugee Women from Iraq and Syria Living in Melbourne, Australia—A Qualitative Study"

_ijerph, 2025, doi:10.3390/ijerph22020149_

Round 1
Reviewer 1 Report
Comments and Suggestions for Authors
This manuscript is well-written but contains a few errors. Overall, the literature gap was clearly identified and justified, and the study's methods produced results that effectively answered the research question.
The questions and comments I included in the table below will help enhance the quality of the literature by providing a deeper insight and fuller understanding of the study conducted and concepts discussed.
Line |
Content |
Reviewer’s comments |
Line 74 |
“…compared to native-born Australian…” |
Does this apply to First Nations women? Can you clarify either way? |
Line 133 |
“Information flyers about…” |
Could you clarify whether the flyers were in English or the preferred language of the women?
|
Line 178 |
“…participant responses in English during the interview or in English…” |
Please clarify. Do you mean “Arabic or in English…”? |
Line 183 |
Reflexive thematic analysis |
Since this analysis method was conducted, I want to know if the interviewer has any reflections on conveying concepts and questions across languages and cultures. How did the interviewer’s perspective influence the questions asked and the responses given? |
Table 1 |
“Country of birth” “Age group” etc. |
Can you differentiate the headings from their subheadings to enhance readability and identification ofy the various demographic categories? |
Line 358 |
“do my own research into it” |
· Did the interviewer explore how participants collect information online? o How do women determine which sources are reliable or not? o Did they encounter conflicting information? If so, how did they determine which information to believe? o Did the participants have information accessible in their language? o Did they conduct the search in English or their language? |
Lines 386-389 |
Despite knowing about it, women did not undertake cervical screening prior to marriage as the belief was it would impact their virgin status and therefore their marriageability “Yes I know about that one [cervical screening] and when I came to Australia I received the mail to do the cervical screening but I didn’t do it because I am a virgin” (Sara, 31) |
How did you conclude the quote to this statement? When reading the quote, it reads as though the participant did not see the need for a cervical screening because she was a virgin rather than being afraid of losing her virginity. Suppose cultural insights had influenced the interpretation of this quote. In that case, the authors should comment on this in the discussion/method as part of their process of conducting reflexive thematic analysis. |
Line 556 |
“Interviewing in English with an Arabic interpreter” |
· Can the authors comment on whether the interview questions and answers had the potential to be misinterpreted through the interpreter? · Does Arabic have equivalent terminologies used by the interviewers? · Did the interpreter/interviewer sense when women didn’t understand what was asked? If yes, how was that addressed? · What qualifications does the interpreter have? Are they part of the research team? |
Line 578 |
“…teach-back method…” |
Considering that some of your participants had limited English and required an interpreter for the interview and their health appointments. Please refer to this paper: Riggs et al. Teach-Back in Interpreter-Mediated Consultations: Reflections from a Case Study. The teach-back method may not be as effective when communication relies on an interpreter.
|
Reviewer 2 Report
Comments and Suggestions for Authors
This article looks at a specific population of refugee women from Iraq and Syria who resettled in Melbourne, Australia, and looks at their perspective on preventing sexual and reproductive health. The study used qualitative semi-structured one-on-one interviews and thematic analysis to better understand the concept. Although this topic has been studied in different refugee populations in different post-structural areas, more information, especially quantitative work, is still needed.
Recommendations for authors:
In the introduction, I suggest providing more information on the target community for your paper rather than general information about global displacement. For example, let us know about the SRH in women from Iraq and Syria before resettlement, their access to preventative care, and their challenges. Also, let us know what other articles have been reporting about these communities. Why is this article important, and what new information will you provide that we still need to read in the other articles?
Also, all citations should be from the original paper rather than a paper that cites your provided information.
At the beginning of the method, instead of a poorly understood phenomenon, I suggest you write your research question clearly.
In the method section, for the study setting, give more information about the target population; what percentage of the population in that area is from Syria or Iraq? What are the characteristics of these women (education, income, marriage, culture, religion)?
For the framework, we need to know more about how this framework impacts your research. Give more information about how you used that, especially on finding your results.
Please provide more information about data analysis steps, coding, and thematic analysis, and the team members who did those.
Davidson's systematic review that you cited as a guide for interview is a complete work in this field; what do you think your work is adding to that?
For limitation and conclusion, you need to mention this is a convenience sampling and qualitative work, and you can not generalize the findings to the total population.
- In line 172, you mentioned face-to-face interviews, but in line 179, you mentioned eight were over the phone.
- It is great to see the table of themes and subthemes, but I suggest identifying the theme based on the research question, which themes relate to health literacy and barriers or enablers.
- In lines 207- 213, you are talking about lack of knowledge vs perception of action. Overall, in this paper, I think you are using the Health Belief model more than the Health Literacy. The interview guide and the analysis are more based on the Health Belief model than the Health Literacy model. You may want to consider changing that or combining the models.
- Lines 386 and 488, the pap smear and even HPV vaccination before marriage are not common in most Muslim-majority countries because the assumption is there is no premarital sex. So, most girls may receive the HPV vaccine around the time of marriage and may have the first pap smear at the time of prenatal care.
- In line 493, your quotes show women's belief that being sick with cancer is in God's hands. It doesn't mention that it impacts their view on screening.
Reviewer 3 Report
Comments and Suggestions for Authors
This research is very important and timely. I thank you the authors for performing this amazing study.
Here are a few comments, I hope the authors find them useful to improve the quality of the paper:
Abstract:
1. Add the study time period in the abstract.
2. How many interviews were conducted?
Introduction:
I would add a paragraph with some statistics on sexual violence in refugee settings in the introduction. The authors briefly covered this topic, but this area needs to be more comprehensive.
Method section:
Include the number of interviews in the methods section; it is already mentioned in the results section.
I see that a volunteer interpreter was involved; did you use a certified volunteer? I want to ensure the quality of the interpretation.
Results:
On page 17, line 423, "To not use it, is for us its 'haram'", please add the country.
Discussion:
Section 6 (Clinical Practice and Research Implications) needs further development. Discuss the six themes and suggest how local and international agencies can improve knowledge and awareness, or add some requirements.
One of the most significant barriers was language, so identifying more GPs who speak the same language is a key finding that needs further explanation.
I have not seen any findings on sexual violence. Did the author ask questions about it? If so, please add the findings; if not, please explain why it was not addressed.
